# Plasma Microbiome in COVID-19 Subjects: An Indicator of Gut Barrier Defects and Dysbiosis

**DOI:** 10.3390/ijms23169141

**Published:** 2022-08-15

**Authors:** Ram Prasad, Michael John Patton, Jason Levi. Floyd, Seth Fortmann, Mariana DuPont, Angela Harbour, Justin Wright, Regina Lamendella, Bruce R. Stevens, Gavin Y. Oudit, Maria B. Grant

**Affiliations:** 1Department of Ophthalmology and Visual Sciences, University of Alabama at Birmingham, 1670 University BLVD, VH490, Birmingham, AL 35294, USA; 2Hugh Kaul Precision Medicine Institute, Department of Medicine, University of Alabama at Birmingham, Birmingham, AL 35294, USA; 3Wright Labs, LLC, Huntingdon, PA 16652, USA; 4Department of Physiology and Functional Genomics, University of Florida, Gainesville, FL 32611, USA; 5Division of Cardiology, Department of Medicine, University of Alberta, Mazankowski Alberta Heart Institute, Edmonton, AB T6G 2B7, Canada

**Keywords:** circulating microbiome, COVID-19, gut barrier permeability, dysbiosis

## Abstract

The gut is a well-established route of infection and target for viral damage by SARS-CoV-2. This is supported by the clinical observation that about half of COVID-19 patients exhibit gastrointestinal (GI) complications. We aimed to investigate whether the analysis of plasma could provide insight into gut barrier dysfunction in patients with COVID-19 infection. Plasma samples of COVID-19 patients (*n* = 146) and healthy individuals (*n* = 47) were collected during hospitalization and routine visits. Plasma microbiome was analyzed using 16S rRNA sequencing and gut permeability markers including fatty acid binding protein 2 (FABP2), peptidoglycan (PGN), and lipopolysaccharide (LPS) in both patient cohorts. Plasma samples of both cohorts contained predominately *Proteobacteria*, *Firmicutes*, *Bacteroides*, and *Actinobacteria*. COVID-19 subjects exhibit significant dysbiosis (*p* = 0.001) of the plasma microbiome with increased abundance of *Actinobacteria* spp. (*p* = 0.0332), decreased abundance of *Bacteroides* spp. (*p* = 0.0003), and an increased *Firmicutes:Bacteroidetes* ratio (*p* = 0.0003) compared to healthy subjects. The concentration of the plasma gut permeability marker FABP2 (*p* = 0.0013) and the gut microbial antigens PGN (*p* < 0.0001) and LPS (*p* = 0.0049) were significantly elevated in COVID-19 patients compared to healthy subjects. These findings support the notion that the intestine may represent a source for bacteremia and contribute to worsening COVID-19 outcomes. Therapies targeting the gut and prevention of gut barrier defects may represent a strategy to improve outcomes in COVID-19 patients.

## 1. Introduction

More than 6.3 million deaths related to COVID-19 have been reported worldwide, a number that is still increasing even after more than 30 months since the diagnosis of the first COVID-19 case [1]. According to a report published by the CDC, in March 2020–July 2022 in the United States, the overall number of COVID-19 cases was higher in female subjects (53.4%) than males (46.6%), while the mortality rate is higher among males (55.1%) than females (44.9%) [2]. The clinical manifestation of COVID-19 is more severe in patients with pre-existing and ongoing medical conditions including cardiovascular diseases, cancer, and diabetes [3,4,5,6,7,8,9,10]. 

Complicating viral pulmonary infections in COVID-19 subjects is the development of secondary bacterial infections, which are fairly frequent in COVID-19 subjects, estimated to be 8.1–14.3% [11,12]. In critically ill patients, this percentage increases up to 34.5%. [13]. Ventilator-associated bacterial pneumonia occurred in 31% of COVID-19 patients who needed ventilation [6], and was associated with substantial mortality in the Wuhan cohort: 28 out of 191 hospitalized patients with COVID-19 developed secondary bacterial infections and all but 1 individual died [6]. Their symptoms of secondary bacterial pneumonia may coincide with those associated with COVID-19, making these infections difficult to diagnose [14]. The source of the secondary infection may be difficult to identify; however, a growing body of evidence suggests that the gut may contribute. 

Support for this comes from the observation that COVID-19 patients experience gastrointestinal (GI) symptoms including nausea, fever, pain, and diarrhea. The most common GI complication is severe diarrhea [15]. During hospitalization, critically ill patients experience GI complications [16]. In a USA-based study, approximately 61.3% of COVID-19 patients reported GI complications, including but not limited to loss of appetite (34.8%), diarrhea (33.7%), mesenteric arterial or venous thromboembolism, and small bowel ischemia [17,18]. These GI complications were associated with longer hospitalization [19]. In a meta-analysis of 107 studies and 15,133 patients combined, the pooled prevalence of GI complications was 10–33.4% [20,21,22]. Although these studies confirm GI findings and important clinical observations, they do not interrogate the pathophysiology associated with these GI complications and whether the gut could be a source of bacteria that can secondarily infect the lung. Thus, we investigated whether COVID-19 patients demonstrated gut barrier defects and presence of a unique microbiome in their plasma. Our patient population was formed of individuals admitted to the University of Alabama at Birmingham hospital (Birmingham, AL, USA) with a confirmed diagnosis of COVID-19, as well as healthy individuals. 

## 2. Results

### 2.1. Clinical Characteristics of the COVID-19 Patients and Healthy Individuals

Out of 146 COVID-19 patients enrolled in the study, the total number of female patients (79; 54.1%) was higher than the male patients (67; 45.9%) (Table 1). At the time of admission to the hospital, all COVID-19 patients were experiencing nausea, myalgia, fever, diarrhea, and shortness of breath. Among the female patients, the quick Sepsis-related Organ Failure Assessment (qSOFA) score showed that 75.9%, 17.7%, and 6.4% of females were classified as having mild, moderate, and severe COVID-19 infection, respectively. In the male patients, the qSOFA score showed 43.3%, 49.3%, 7.4% as mild, moderate, and severe infection, respectively. Based on the severity of these symptoms and duration of the recovery period, the length of the hospitalization varied from 1–123 days. Among those with mild, moderate, and severe COVID-19 infection, diabetic comorbidities were present in 33.7%, 38.3%, and 30% of subjects, respectively, and in-hospital mortality was 1.1%, 8.5%, and 60%, respectively. Additionally, other comorbidities such as cardiac, pulmonary, and oncologic issues were also present in 16.43%, 23.97, and 13.01% of COVID-19 patients, respectively. 

### 2.2. Laboratory Findings and COVID-19 Manifestation in Patients

Laboratory observations for the COVID-19 cohort included a metabolic panel (Table 2) and differential complete blood count (Table 3). Of COVID-19 positive subjects, C-reactive protein (CRP, *q* < 0.0001), and procalcitonin (*q* = 0.035) were found to be elevated in subjects with greater COVID-19 severity; whereas no statistically significant changes were observed in ferritin, hemoglobin, glucose, D-dimer, Hs troponin-I, BNP, LDH, and lactate between cohorts based on COVID-19 severity (Table 2). 

CBC results indicated an increase in circulating white blood cells (WBC, *q* = 0.004), a decrease in circulating lymphocytes (*q* = 0.028), and an increase in circulating neutrophils (*q* = 0.041) in COVID-19 subjects with increased severity. We did not find statistically significant alterations in red blood cell count, platelet count, monocytes, basophils, and eosinophils among cohorts based on COVID-19 severity (Table 3).

### 2.3. Presence of Gut Microbial Abundance in the Blood of COVID-19 Patients

Plasma samples from COVID-19-positive and healthy individuals were obtained under sterile conditions and evaluated for the presence and structure of bacterial communities by 16S rRNA sequencing. We obtained a total of 287,656 sequencing reads between all 36 subjects (15 COVID-19-positive and 21 healthy individuals); there were no alterations in read counts between healthy control and COVID-19-positive subjects (*p* = 0.5383, Figure 1A). In assessing the alpha (α) diversity, a representation of the total microbial population within a single sample, we utilized the following measures/indices: Faith’s Phylogenetic Diversity Index (DI), observed operational taxonomic units (OTUs), Shannon’s DI, and Pielou’s Evenness. We found that there were no statistically significant alterations in Faith’s Phylogenetic Diversity (*p* = 0.2915), Observed Operational Taxonomic Units (OTUs, *p* = 0.7387), or Shannon’s Diversity Index [23,24] (*p* = 0.121) between healthy control subjects and those with COVID-19 infection; however, there was a trend in the reduction of alpha diversity (Faith’s Phylogenetic and Shannon’s Diversity Index). A statistically significant decrease in the score of Pielou’s Evenness was observed in COVID-19 patients compared to healthy individuals [23,25] (*p* = 0.0156) (Figure 1B–E). These data indicate there was a decrease in mean evenness within the circulating microbiota distribution, though there were no aggregate differences in other measures of α-diversity. Beta (β)-diversity was visualized using Principal Coordinate Analysis (PCoA) and indicated differential clustering of healthy and COVID-19 positive microbiomes (*p* = 0.001, Figure 1F). Thus, we next aimed to determine which taxa are contributing to the altered circulatory microbiome of COVID-19 subjects.

### 2.4. Phylogenic Differences in Plasma Microbiome in the COVID-19 Plasma Samples

The relative abundance of microbial composition in COVID-19 samples is shown in Figure 2A. Four major phyla were identified in the plasma of healthy and COVID-19-positive subjects: *Actinobacteria*, *Bacteroides*, *Firmicutes*, and *Proteobacteria*. We observed an enrichment in the abundance of *Actinobacteria* (*p* = 0.0332) and a decrease in abundance of *Bacteroidota* (*p* = 0.0003) in COVID-19-infected subjects compared to healthy control subjects (Figure 2B,C). We did not find any change in abundance of *Firmicutes* or *Proteobacteria* between cohorts (Figure 2D,E). However, we did find that the *Firmicutes:Bacteroidetes* (F:B) ratio, a gross determinant of microbiota composition which is well-known to increase in pathological conditions including COVID-19 [26,27,28,29,30,31], was increased (*p* = 0.0003) in COVID-19 subjects compared to healthy control subjects (Figure 2F). We observed that 6–7 out of 10 moderate COVID-19 patients (Table 4) show greater phylogenic differences when looking at an individual sample (Figure 2B–E); however, this observation is based on a small sample size. 

Next, the abundance of each microbial population was assessed, which revealed that, at the genus level (Figure 3), the prevalence of *Aquabacterium*, *Brevibacterium*, *Pantoea*, *and Sphingobacterium* were enriched in the plasma of COVID-19 subjects, whereas COVID-19-positive subjects exhibited a decrease in *Streptococcus*, *Prevotella*, *Haemophilus*, *Gemella*, *Actinomyces*, *Lachnospiraceae*, and *Bifidobacterium* compared to healthy control subjects. These data support the notion that COVID-19-positive subjects demonstrate a decrease in classically beneficial bacteria, primarily *Bifidobacterium* [32,33,34,35] and an increase in bacteria which have been shown to be associated with bacteremia/sepsis in humans, primarily *Brevibacterium* [36,37,38,39,40] and *Pantoea* [41,42] species. Together, these data support a potentially clinically relevant enrichment in pathogenic bacteria and a loss of bacteria that support gut health in the circulation of COVID-19-positive subjects. 

### 2.5. SARS-CoV-2 Infections Promote Gut Barrier Defects and Endotoxemia in COVID-19 Patients

The plasma microbiome arises largely because of bacterial translocation from the gut into the systemic circulation [43,44,45,46,47,48]. Compromised intestinal barriers are an important pathogenic factor and contribute to promotion of inflammation. We measured gut permeability markers in the plasma of COVID-19 and control subjects. FABP2 is an intracellular protein which is expressed specifically in intestinal epithelial cells [49] and binds free fatty acids, cholesterol, and retinoids, and is involved in intracellular lipid transport. During mucosal damage, mature epithelial cells release this protein into the circulation [50] and higher levels of FABP2 in the plasma are associated with gut barrier defects [49,51,52,53]. To determine the integrity of the gut barrier in COVID-19 patients, the levels of plasma FABP2 were measured. As seen in Figure 4A, the levels of FABP2 were higher (*p* = 0.0013) in the plasma of COVID-19 patients compared with healthy individuals, supporting the assertion that subjects with COVID-19 infection exhibit increased permeability of the gut barrier. To assess this gut barrier defect in the context of endotoxemia, we next wanted to determine the serum abundance of gut microbial antigens (GMAs), bacterial ligands which are released into the circulation from the intestinal lumen with a dysfunction gut barrier.

GMAs initiate deleterious signaling pathways and contribute to systemic inflammation [43,54,55,56,57,58,59,60,61,62,63]. To determine if gut barrier dysfunction led to translocation of GMAs into the circulation of COVID-19-positive patients, we measured abundance of plasma PGN and LPS, the major cell wall components of Gram-negative and Gram-positive bacteria, respectively, which are known to increase in the circulation with gut barrier disruption. The levels of PGN were increased nearly 2.5 times (*p* < 0.0001) in COVID-19-positive patients compared with controls (Figure 4B). LPS (*p* = 0.005) was found in higher levels in COVID-19 samples compared with non-COVID-19-infected patients (Figure 4C). Together, these data support the notion that subjects with COVID-19 exhibit increased permeability for the gut barrier which results in increased translocation of microbial antigens into the circulation, where they play a role in augmenting systemic inflammation and COVID-19 pathogenesis. We next aimed to determine if increased circulating immune cells and endotoxemia in COVID-19-positive subjects was associated with altered abundance of circulating inflammatory mediators.

### 2.6. COVID-19 Infections Promote Increased Pro-Inflammatory Cytokine Production

We assessed the circulating inflammatory milieu of COVID-19-positive subjects to elucidate any cytokines/chemokines which may be responsive to endotoxemia present in these patients. As can be seen in Table 5, we found that the plasma concentration of the proinflammatory IL-1β (*p* = 0.011) to be decreased in the plasma of COVID-19-positve subjects; whereas, IL-6 (*p* = 0.03), IL-8 (*p* = 0.002), IFN-γ (*p* = 0.03), TNF-α (*p* = 0.006), MCP-1 (*p* < 0.0001), MIP-1α (*p* = 0.034), and MIP-1β (*p* = 0.049) were increased in COVID-19-positive subject plasma. We did not find a difference in the plasma abundance of IL-2 (*p* = 0.28), IL-12p70 (*p* = 0.54), IL-17 (*p* = 0.95), and GM-CSF (*p* = 0.35) in COVID-19-positive subjects compared to healthy control subjects (Table 5). In assessing the abundance of anti-inflammatory cytokines in the plasma of COVID-19-positive subjects, we found the concentration of IL-5 (*p* = 0.04) to be decreased and IL-10 (*p* = 0.036) to be increased compared to healthy control subjects; no difference was found in the concentration of IL-4 (*p* = 0.75), IL-9 (*p* = 0.22), and IL-13 (*p* = 0.84) in COVID-19-infected subject plasma (Table 5). Together, these data demonstrate a substantial increase in proinflammatory cytokines and moderate alterations in anti-inflammatory cytokine abundance in the plasma of patients with active COVID-19 infection compared to those without. These data support the notion that increased gut permeability and circulatory microbiota dysbiosis may be pathologically driving an increased proinflammatory response in subjects with active COVID-19 infection. 

## 3. Discussion

In recent years, the notion of a circulating microbiota which changes in composition during pathological states has gained increasing support [64,65,66]. We postulate that an altered circulating microbiota in COVID-19-positive subjects may participate in exacerbation of pathology and increase the likelihood for systemic bacterial infection [67]. While most studies to date examine the blood metabolome, rather than the blood microbiome, we first sought to establish whether a unique plasma microbiome existed in COVID-19-infected subjects and then determine if the microbial diversity supported that the origin of these microbes was the intestine [68,69]. Results from numerous studies have linked the plasma metabolome to the gut microbiome and their implication for specific diseases [70]. Specifically, in agreement with our previous study [71] and the current study, Venzon et al. also showed plasma dysbiosis and increased gut barrier leakage in COVID-19-positive subjects [72]. Wikoff et al. demonstrated that the gut microbiome dramatically influenced the composition of blood metabolites using MS-based methods and plasma extracts from germ-free mice compared with samples from conventional animals [73]. Bacterial-mediated production of bioactive indole-containing metabolites derived from tryptophan such as indoxyl sulfate and the antioxidant indole-3-propionic acid (IPA) have been identified in the plasma. 

The fecal microbiome also has been compared to the plasma metabolome in disease states such as ulcerative colitis, where products of sphingolipid metabolism, specifically sphingosine 1-phosphate in the blood correlate with *Roseburia*, *Klebsiella*, and *Escherichia-Shigella* [74]. Kurilshikov et al. showed gut microbiome explained up to 16% of the variation in 231 major plasma metabolites [75], highlighting its powerful impact on the host and the multidimensional interplay between gut bacteria and their ability to predict human disease or health. However, studies on the plasma microbiome are limited. 

Here, we demonstrated an enrichment of *Actinobacteria and Firmicutes* and a depletion of *Bacteroidota* and *Proteobacteria* at phylum level in the plasma of COVID-19-infected patients compared to healthy controls. Studies have been suggested that an increase in the abundance of *Actinobacteria* and *Firmicutes* and reduction in *Bacteroidota* and *Proteobacteria* trigger chronic respiratory diseases, including asthma and chronic obstructive pulmonary disease (COPD), as well as respiratory virus infection in smokers than non-smokers [76,77]. COPD enhances the risk of serious illness in COVID-19 patients [78,79]. Whittle et al. performed a comprehensive evaluation of the blood microbiome in healthy and asthmatic individuals and found, at the phylum level, the blood microbiome was predominately composed of *Proteobacteria*, *Actinobacteria*, *Firmicutes*, and *Bacteroidetes* [64]. These key phyla detected were consistent irrespective of the molecular method used for their identification (DNA vs. RNA) and were consistent with the results of other published studies [80,81,82,83]. 

Studies by Serena et al. demonstrate that celiac disease patients exhibit alterations in blood microbiome composition and taxonomic diversity compared to healthy subjects and they suggested that changes in the blood microbiome may contribute to the pathogenesis of celiac disease [84]. Buford et al. compared microbiota profiles of serum from healthy young (20–35 years) and older adults (60–75 years). They demonstrated that the richness and composition of the serum microbiome differ between these age groups and are linked to indices of age-related inflammation such as IL-6 and TNFα [85]. 

During hospitalization, the fecal microbiome can be altered; thus, we selected to evaluate the initial plasma samples of COVID-19-infected patients. In a small group of 15 patients, depletion of the commensal bacterium *Lactobacillus* was documented in 65% of patients during COVID-19 infection. Commensal bacteria act on the host’s immune system to induce a protective response and also inhibit the growth of respiratory pathogens [86]. Heeney et al. reported reduced abundance of *Lactobacillus* in diabetes, obesity, and cancer [87]. We did not find a difference in presence of *Lactobacillus* species within the plasma of COVID-19 subjects, though we did find a decrease in dairy-derived *Bifidobacterium* in the plasma of subjects with COVID-19. *Bifidobacterium* spp. are a well-characterized family of bacteria that are homeostatically present in the gut microbiota of humans and animals [88,89,90]. Decreased abundance of *Bifidobacterium* spp. in the gut microbiota of humans has been associated with occurrence of colorectal cancer [91], inflammatory bowel disease [92], diabetes [93], and obesity [94]. It is entirely plausible that the depletion of *Bifidobacterium* spp. in the gut microbiota of COVID-19-positive subjects participates in the development and exacerbation of gastrointestinal symptoms in these patients. 

In addition, we found there to be an increase in the abundance of several genera which are associated with bacteremia and sepsis in human subjects, *Brevibacterium* and *Pantoea.* Sepsis is defined as a life-threatening condition in which the body’s immune system damages its own tissues in response to infections [95]. Alhazzani et al. reported that most COVID-19 related deaths are caused by sepsis [96]. Even after viral clearing, there was a loss of salutary species in most COVID-19-positive patients, suggesting that exposure to SARS-CoV-2 might be associated with more long-lasting deleterious effects on the gut microbiome. 

Due to their role in regulating immune function and metabolism, gut microbes are key contributors in the normal physiology [97,98,99,100]. The fecal microbiota and its translocation from the gastrointestinal tract into systemic circulation has been considered as a key driver of immune response and systemic inflammation [101,102,103,104]. Abnormal presence of gut microbes in the plasma can initiate and intensify inflammatory cascades [105]. Although systemic and local tissue inflammation is paramount in the pathogenesis of COVID-19 infection, the clinical relevance of gut microbes in the plasma remains unclear. Therefore, in this study we sought to evaluate the hypothesis that bacterial translocation from the intestine into the systemic circulation occurs and is associated with worsened outcomes in SARS-CoV-2 infection. Increased intestinal permeability due to mucosal barrier dysfunction could result in microbial translocation. Our results support that the COVID-19-positve patients exhibit gut barrier dysfunction as evidenced by the higher levels of FABP2, PGN, and LPS and the abnormal presence of microbes in their plasma. 

In addition, we found that the circulating inflammatory milieu of COVID-19-positive subjects was enriched in abundance of white blood cells, lymphocytes, and neutrophils compared to healthy subjects. Concomitantly, we found those with COVID-19 infection exhibit increased circulatory proinflammatory cytokines (IL-6, IL-8, IFN-γ, TNF-α, MCP-1, MIP-1α, MIP-1β) and moderate changes in abundance of anti-inflammatory cytokines (IL-5 and IL-10). Together, these studies support the growing literature showing that COVID-19 subjects exhibit a proinflammatory immune profile which is associated with increased gut permeability, endotoxemia, and dysbiosis of the circulatory microbiome.

Thus, our studies provide evidence for the loss of gut barrier function in COVID-19-positive subjects; however, the mechanisms responsible have not been elucidated. 

This study has limitations due to limited sample availability, including the inability to examine the plasma microbiome and markers of gut permeability and endotoxemia of our entire COVID-19-positive cohort. Due to limitation in the amount of plasma sample we could obtain per patient, we could only either perform 16S analysis or measurement of gut leakage markers. Therefore, our observations are not suitable for correlation analysis. 

Despite these limitations, we show conclusively that gut barrier leakage occurs in COVID-19-positive subjects. Taken together, we show the presence of potentially pathogenic bacteria in the plasma of COVID-19-positive subjects which is associated with disruption of the gut barrier and elevation of systemic bacterial LPS and PGN which serve to enhance systemic inflammation (Figure 5). Therefore, leaky gut and microbial dysbiosis may contribute to cytokine storm in patients severely ill with COVID-19. 

## 4. Material and Methods

### 4.1. Study Subjects

A total of 146 COVID-19-positive patients participated in this study. During hospitalization of the COVID-19-positive patients at UAB hospital, blood samples were collected within 48 hrs of their admission, under sterile conditions following Institutional Review Board guidelines. Blood samples from healthy individuals were collected following routine guidelines [106]. Using the quick Sepsis-related Organ Failure Assessment (qSOFA) as a guide to COVID-19 severity, COVID-19-positive subjects were classified as mild (qSOFA = 0), moderate (qSOFA = 1–2), and severe (qSOFA = 3). Patient characteristics and demographics can be found in Table 1. From these patients, a total of 15 COVID-19-positive subjects and 17 healthy individuals were selected to undergo plasma microbiome assessment.

### 4.2. Microbial DNA Extraction and 16S rRNA Sequencing

The frozen plasma samples were shipped to Wright Labs, LLC. for 16S rRNA sequencing (V3–V4 region). Microbial DNA was extracted from samples using the DNA/RNA Miniprep Kit (Zymo Research, Irvine, CA, USA) according to the manufacturer’s instructions. After extraction, DNA purity and concentration were determined using Qubit 4 Fluorometer (Invitrogen, Carlsbad, CA, USA) and dsDNA HS assay kit (ThermoFisher Scientific, Waltham, MA, USA). PCR products were pooled, and gel-purified on a 2% agarose gel using Qiagen Gel Purification Kit (Qiagen, Frederick, MD, USA). After a quality check using 2100 Bioanalyzer and DNA 1000 chip (Agilent Technologies, Santa Clara, CA, USA), 16S rRNA sequencing was performed using an Illumina MiSeq v2 chemistry with paired-end 250 base pair reads as per the Earth Microbiome Project’s protocol [107]. One negative control was processed in parallel with the samples and sequenced as well.

### 4.3. Bioinformatic Analysis

Raw sequence data was successfully obtained and imported into Qiime2 for processing and analyses [108]. Initial quality in the form of Phred q scores was determined using Qiime2, while cumulative expected error for each position was determined with VSEARCH [109]. Based on these quality data, forward and reverse reads were truncated at a length of 250, with a maximum expected error of 0.5 within Qiime2’s implementation of the DADA2 pipeline [110]. Qiime2’s DADA2 pipeline was also used to merge forward and reverse reads and removed chimeras and assign the remaining sequences to amplicon sequence variants (ASVs). Representative sequences were used to determine taxonomic information. The full report and statistical analyses from Wright Labs, Huntingdon, Pennsylvania are available upon request.

### 4.4. Alpha and Beta Diversity Analysis

Alpha diversity was calculated by subsampling the ASV table at 10 different depths, ranging from 230 to 2300 sequences, for the Faith’s Phylogenetic Diversity [111], Observed OTUs [112], Pielou’s Evenness [113], and Shannon’s Index [114] metrics. In total, 20 iterations were performed at each depth to obtain average alpha diversity values for the different metrics. A rarefaction plot was created with the results of this subsampling to confirm that the diversity approached an asymptote and the slope decreased as depth increased. Averages for the greatest depth were calculated and plotted to show each sample’s diversity. 

Beta diversity analyses were conducted after the ASV table had first undergone cumulative sum scaling normalization [115] to mitigate differences between samples based on sequencing depth. Distances between samples were calculated using the Weighted Unifrac metric [116] based on the normalized table and rooted tree. The resulting distance matrix was visualized as a Principal Coordinates Analysis plot in R. 

### 4.5. Measurement of Gut Permeability Marker FABP2

The level of FABP2 [51], a marker of intestinal barrier damage, was determined by ELISA in the plasma samples using a colorimetric assay kit (#DFBP20, R&D systems, Minneapolis, MN, USA) following the manufacturer’s protocol. The absorbance was measured at 450 nm using a microplate reader, and the levels of FABP2 were calculated as per the standard curve and expressed as pg/mL.

### 4.6. Enzyme-Linked Immunosorbent Assay for Measuring Gut Microbial Peptide Translocation into the Systemic Circulation

The level of PGN in plasma samples was measured using a colorimetric assay kit for human PGN (#MBS261545, MyBioSource Inc., San Diego, CA, USA) following the manufacturer’s protocol. The absorbance was measured at 450 nm using a microplate reader and the levels of peptidoglycans were calculated as per the standard curve and expressed as ng/mL. The levels of LPS were also measured by ELISA kit (#EKC34448, Biomatik, Wilmington, DE, USA) following the manufacturer’s instruction manual. The levels of LPS were calculated by standard curve and expressed as pg/mL.

### 4.7. Immunological Marker Detection in Human Plasma

Plasma levels of immunoregulatory cytokines GM-CSF, IGN-γ, IL-1β, IL-2, IL-4, IL-5, IL-6, IL-8, IL-9, IL-10, IL-12p70, IL-13, IL-17A, MCP1, MIP1α, MIP1β, and TNF-α were measured by using an advanced particle-based flow cytometry approach following the manufacturer’s instructions. The human immunoassay panel (FirePlex^®^: ab243549, Waltham, MA, USA) for key cytokines was purchased from Abcam (Waltham, MA, USA). Briefly, capture particle solution was added to the 96-well filter plate. The solution was then removed from the filter plate by using a vacuum manifold (ab204067, Waltham, MA, USA). To each well, 50 µL of plasma samples (4-fold dilution)/standards were added, followed by overnight incubation at 4 °C with orbital shaking at 750 rpm. After overnight incubation, the filter plate was washed by applying a gentle vacuum. Biotin-conjugated antibody was then added to each well and incubated for one hour at room temperature with orbital shaking at 750 rpm followed by washing. Samples were then incubated with reporter solution for 30 min at room temperature with orbital shaking at 750 rpm followed. After careful washing, 175 µL of FirePlex cytometry running buffer-I (ab245836, Waltham, MA, USA) was added to each well, and samples were acquired using a BD FACSymphony A5 cell analyzer (Franklin Lakes, NJ) equipped with a 96-well plate high-throughput sampler. Data were analyzed using FirePlex Analysis Workbench software provided by Abcam and presented as pg/mL.

### 4.8. Statistical Analysis

Data were evaluated for presence of outliers and adherence to a normal distribution using GraphPad Prism software (San Diego, CA, USA), version 8.1. Statistical significance of normally and non-normally distributed data were assessed via Student’s *t*-test and the Mann–Whitney U test, respectively, with α = 0.05. 

## Figures and Tables

**Figure 1 ijms-23-09141-f001:**
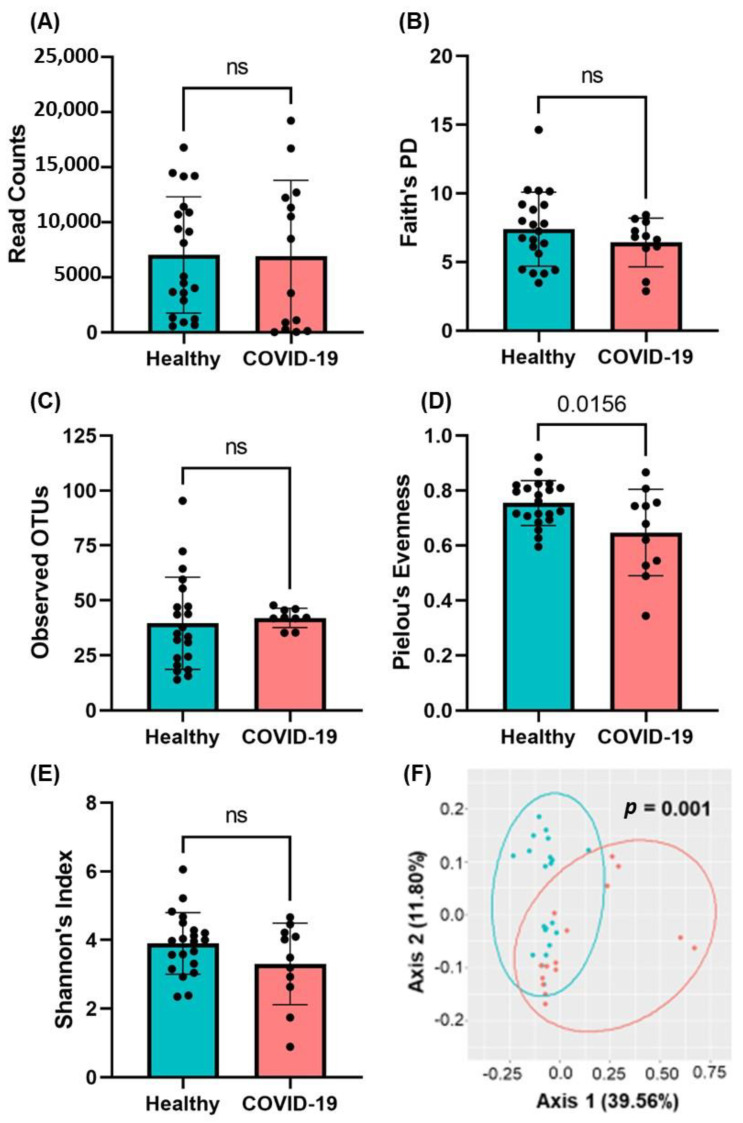
16S rRNA analyses in the plasma of COVID-19 patients. (**A**) Total read counts and measures of alpha diversity—(**B**) Faith’s Phylogenetic Diversity, (**C**) Observed OTUs, (**D**) Pielou’s Evenness, and (**E**) Shannon’s Dysbiosis Index—were calculated and indicate no aggregate changes in alpha diversity. Data are presented as mean ± S.E.M. Each dot represents a sample in the cohorts. Student’s *t*-test *p*-values are indicated where applicable (ns, *p* > 0.05). (**F**) Unsupervised 2D principal coordinates analysis (PCoA) of weighted UniFrac distance revealed significant alterations in beta-diversity in COVID-19 subjects (PERMANOVA *p*-value). Each dot indicates one patient plasma sample.

**Figure 2 ijms-23-09141-f002:**
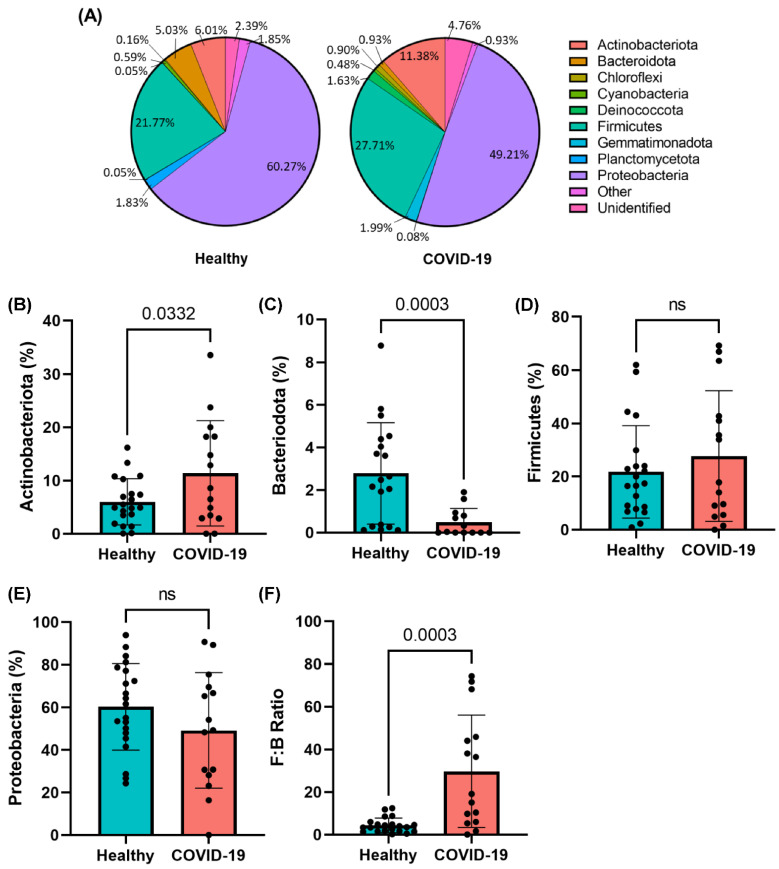
COVID-19-positive subjects exhibit significant dysbiosis of the plasma microbiome at the phylum level. (**A**) Pie charts representing dominant phyla that constitute the circulating microbiome in COVID-19-positive subjects. Individual phyla (**B**–**E**) which were found to be differentially abundant in the plasma of COVID-19-positive subjects including increased *Actinobacteria* and decreased *Bacteroidota*. (**F**) The *Firmicutes:Bacteroidetes* (F:B) ratio indicates a significant increase in dysbiosis of the dominant phyla. Data are presented as mean ± S.E.M. Each dot represents a sample in the cohorts. Student’s t-test *p*-values are indicated where applicable (ns, *p* > 0.05).

**Figure 3 ijms-23-09141-f003:**
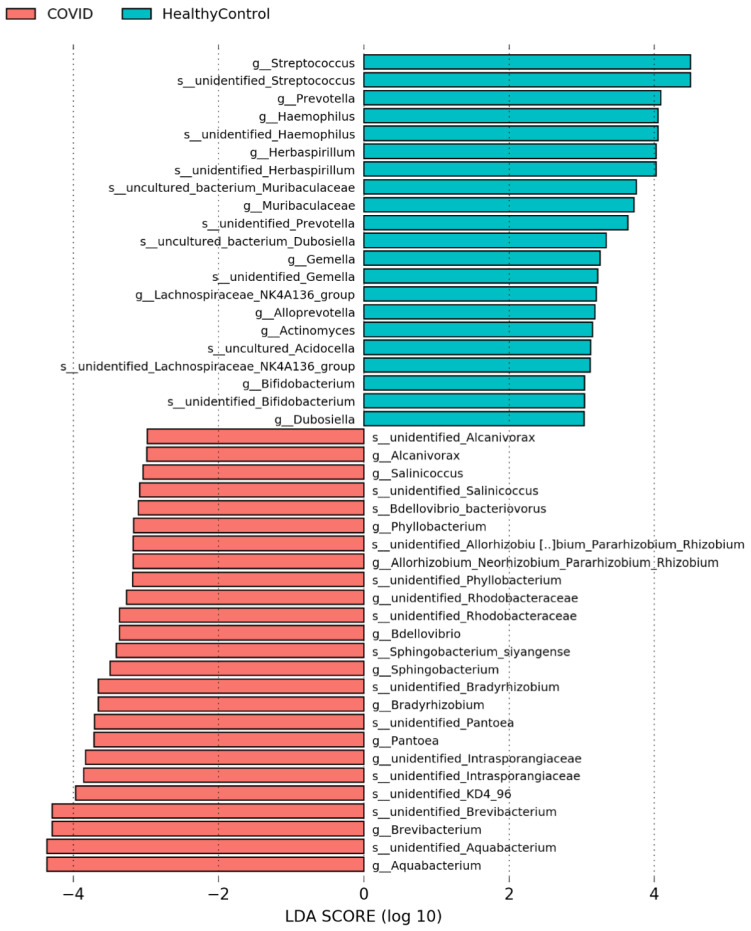
COVID-19-positive subjects exhibit significant dysbiosis of the plasma microbiome at the genus/species level. Linear discriminant analysis (LDA) of CPM normalized counts of Metaphlan displaying differential abundances of several prominent genera in the COVID-19 plasma samples including *Bifidobacterium*, *Pantoea*, *Streptococcus*, and *Brevibacterium* spp.

**Figure 4 ijms-23-09141-f004:**
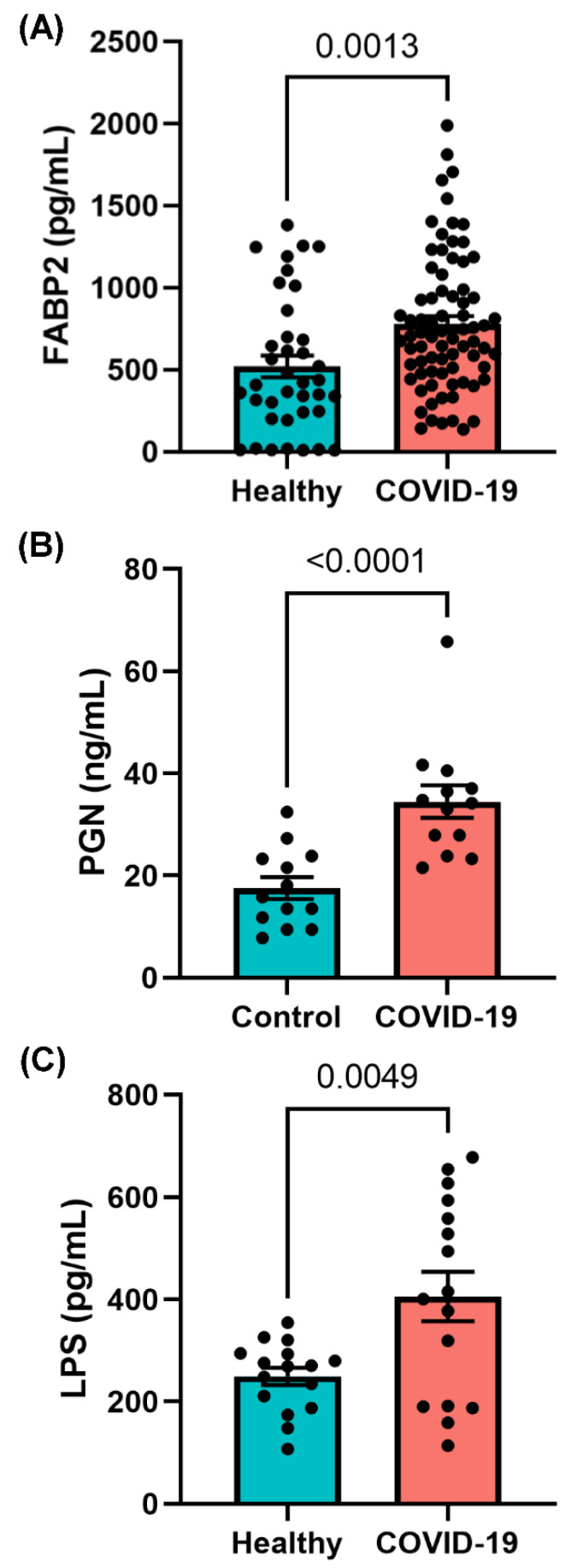
SARS-CoV-2 infection induces increased gut permeability and endotoxemia. ELISA results of plasma (**A**) fatty acid binding protein 2 (FABP2) and the gut microbial antigens (**B**) peptidoglycan (PGN) and (**C**) lipopolysaccharide (LPS) which indicate increased gut barrier integrity and increased endotoxemia in COVID-19-infected subjects. Data are presented as mean ± S.E.M. Each dot represents a sample in the cohorts. Student’s *t*-test *p*-values are indicated where applicable.

**Figure 5 ijms-23-09141-f005:**
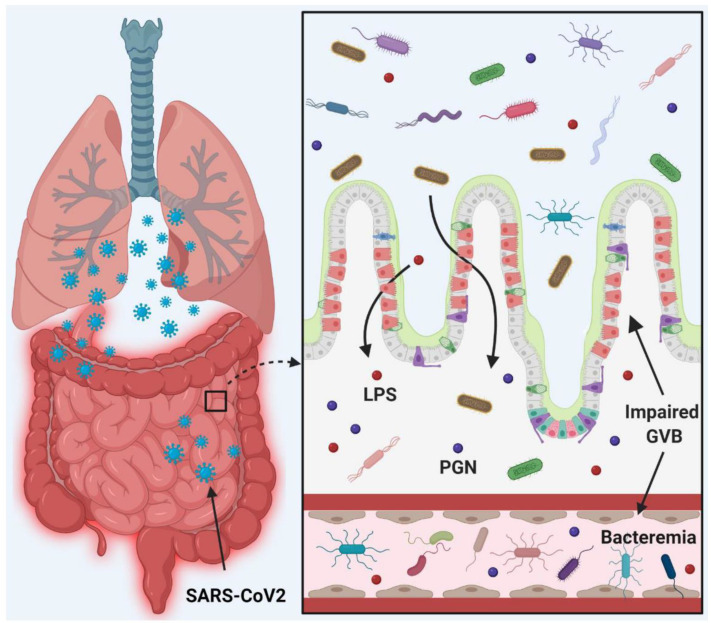
Schematic diagram representing the hypothesis of COVID-19 infection promoting gut barrier defects and translocation of the gut microbiome into the systemic circulation, resulting in worsened outcomes. This figure was generated via Biorender.com.

**Table 1 ijms-23-09141-t001:** Characteristics, demographics, and clinical observations of COVID-19 patients.

Patient Characteristics	N ^1^	Severity on Admission			*p*-Value ^2^
		(qSOFA 0) Mild (N = 89)	(qSOFA 1–2) Moderate (N = 47)	(qSOFA 3) Severe (N = 10)	
Sex, *n* (%)	146				<0.001
Female		60 (67)	14 (30)	5 (50)	
Male		29 (33)	33 (70)	5 (50)	
Age Range (Years), *n* (%)	144				0.65
<50		16 (18)	10 (22)	3 (30)	
>75		6 (6.8)	5 (11)	1 (10)	
50–75		66 (75)	31 (67)	6 (60)	
Unknown		1	1	0	
Diabetic Comorbidity, *n* (%)	114				0.36
Diabetes History		30 (40)	18 (55)	3 (50)	
No Diabetes History		45 (60)	15 (45)	3 (50)	
Unknown		14	14	4	
Cardiac Comorbidity, *n* (%)	114				0.012
Heart Failure or MI History		10 (13)	11 (33)	3 (50)	
No Cardiac History		65 (87)	22 (67)	3 (50)	
Unknown		14	14	4	
Pulmonary Comorbidity, *n* (%)	114				0.25
COPD History		23 (31)	12 (36)	0 (0)	
No COPD History		52 (69)	21 (64)	6 (100)	
Unknown		14	14	4	
Oncologic Comorbidity, *n* (%)	114				0.83
Cancer or Metastatic Tumor History		13 (17)	6 (18)	0 (0)	
No Oncologic History		62 (83)	27 (82)	6 (100)	
Unknown		14	14	4	
Outcomes					
Hospitilization (days), *n* (%)	145				0.022
<15		81 (92)	40 (85)	7 (70)	
>30		5 (5.7)	2 (4.3)	0 (0)	
16–30		2 (2.3)	5 (11)	3 (30)	
Unknown		1	0	0	
In-Hospital Mortality, *n* (%)	146				<0.001
Deceased In-Hospital		1 (1.1)	4 (8.5)	6 (60)	
Discharged		88 (99)	43 (91)	4 (40)	
ICU Admission (anytime), *n* (%)	146				<0.001
ICU Admission		2 (2.2)	12 (26)	9 (90)	
No ICU Admission		87 (98)	35 (74)	1 (10)	
Vasopressor Therapy, *n* (%)	146				<0.001
Required Vasopressor		3 (3.4)	6 (13)	9 (90)	
No Vasopressor		86 (97)	41 (87)	1 (10)	
Invasive Mechanical Veniltaion, *n* (%)	146				<0.001
Required Ventilation		3 (3.4)	5 (11)	9 (90)	
No Ventilation		86 (97)	42 (89)	1 (10)	
Continuous Renal Replacement Therapy, *n* (%)	146				<0.001
Required CRRT		0 (0)	2 (4.3)	4 (40)	
No CRRT COVID-19+ Evidence		89 (100)	45 (96)	6 (60)	
COVID-19 Billing Code Evidence, *n* (%)	146				<0.001
COVID19+ Billing Code During Encounter		58 (65)	47 (100)	10 (100)	
No Billing Code		31 (35)	0 (0)	0 (0)	
COVID-19 Laboratory Test Evidence, *n* (%)	146				<0.001
COVID19+ Test During Encounter		18 (20)	47 (100)	10 (100)	
No Test		71 (80)	0 (0)	0 (0)	

^1^ N represents distinct encounters, ^2^ Fisher’s exact test.

**Table 2 ijms-23-09141-t002:** Laboratory findings of COVID-19 patients.

Laboratory Value ^1^	N ^2^	Severity on Admission			*p*-Value ^3^	*q*-Value ^4^
		(qSOFA 0) Mild (N = 59)	(qSOFA 1–2) Moderate (N = 4)	(qSOFA 3) Severe (N = 10)		
Ferritin (ng/L)	65	326 (196–1184)	478 (188–1065)	382 (305–1665)	0.83	>0.99
C-reactive Protein (mg/L)	100	26 (7–84)	90 (45–131)	136 (43–172)	<0.001	<0.001
Hemoglobin (g/dL)	116	13.40 (11.85–14.55)	12.35 (11.15–13.83)	10.05 (9.03–11.86)	0.006	0.059
Glucose (mg/dL)	116	114 (100–142)	122 (110–154)	150 (128–189)	0.040	0.40
D-Dimer (mg/L FEU)	96	287 (218–551)	451 (300–1324)	556 (435–920)	0.008	0.076
Procalcitonin (ng/mL)	65	0.07 (0.05–0.09)	0.12 (0.07–0.47)	0.77 (0.12–3.00)	0.004	0.035
Hs Troponin-I (ng/L)	60	8 (5–13)	10 (5–31)	20 (8–33)	0.32	>0.99
BNP (pg/mL)	39	103 (68–137)	76 (25–120)	90 (66–180)	0.36	>0.99

^1^ Data presented are laboratory values collected within 3 days of admission and are displayed as median (IQR) unless otherwise indicated. ^2^ N represents distinct encounters, ^3^ Kruskal-Wallis rank sum test, ^4^ Bonferroni correction for multiple testing.

**Table 3 ijms-23-09141-t003:** Immunological features of COVID-19 patients.

Laboratory Value ^1^	N ^2^	Severity on Admission			*p*-Value ^3^	*q*-Value ^4^
		(qSOFA 0) Mild (N = 58)	(qSOFA 1–2) Moderate (N = 50)	(qSOFA 3) Severe (N = 12)		
Red Blood Cell Count (×10^3^/uL)	120	4.56 (4.12–4.87)	4.63 (4.31–5.06)	4.52 (4.08–4.77)	0.45	>0.99
Platelet Count (×10^3^/uL)	120	212 (170–260)	225 (160–282)	228 (161–281)	>0.99	>0.99
White Blood Cell Count (×10^3^/uL)	120	5.7 (4.0–8.2)	8.4 (6.2–11.2)	9.8 (5.7–12.7)	<0.001	0.004
Lymphocytes (relative; %)	117	21 (11–37)	12 (8–20)	10 (7–21)	0.003	0.028
Neutrophils (relative; %)	117	68 (52–79)	76 (68–86)	78 (72–88)	0.005	0.041
Monocytes (relative; %)	117	8.0 (6.0–10.0)	8.0 (5.2–11.8)	6.0 (5.0–8.8)	0.22	>0.99
Basophils (relative; %)	116	1.00 (0.00–1.00)	0.00 (0.00–1.00)	0.00 (0.00–0.75)	0.046	0.36
Eosinophils (relative; %)	84	1.00 (0.00–2.75)	0.00 (0.00–1.25)	1.00 (0.00–2.00)	0.073	0.59

^1^ Data presented are laboratory values collected within 3 days of admission and are displayed as median (IQR) unless otherwise indicated. ^2^ N represents distinct encounters, ^3^ Kruskal-Wallis rank sum test, ^4^ Bonferroni correction for multiple testing.

**Table 4 ijms-23-09141-t004:** Characteristics of COVID-19 patients assessed for plasma microbiome.

Patient’s Charactersitics	Severity on Admission	
	Mild	Moderate
Total number, *n* = 15 (%)	5 (33.3%)	10 (66.6%)
Sex, *n* (%)		
Male	4 (40%)	6 (60%)
Female	1 (20%)	4 (80%)
Age Range (Years), *n* (%)		
<30	0	0
30–50	2 (40%)	2 (20%)
>50	3 (60%)	8 (80%)
Diabetes, *n* (%)	3 (60%)	2 (20%)
Thrombosis, *n* (%)	1 (20%)	7 (70%)
Hospitalization, *n* (%)		
<15	3 (60%)	4 (40%)
16–30	1 (20%)	5 (50%)
>30	1 (20%)	1 (10%)
Mortality, *n* (%)	0	5 (50%)
Diabetic patients with COVID-19, *n* (%)	0	0
Thrombosis in COVID-19 patients, *n* (%)	0	5 (50%)

N represents distinct encounters. The percentage of data was calculated for total number of individuals in the category (mild/moderate).

**Table 5 ijms-23-09141-t005:** Plasma cytokine/chemokine expression in COVID-19 and healthy subjects.

Cytokine/Chemokine	Mean ± SD (pg/mL)		95% CI (pg/mL)		*p*-Value
	Healthy	COVID-19	Healthy	COVID-19	
** *Pro-inflammatory* **					
**IL-1β**	1.46 ± 0.61	0.96 ± 0.57	(1.16, 1.77)	(0.71, 1.21)	0.0106 *
**IL-2**	3.21 ± 1.26	2.75 ± 1.34	(2.58, 3.84)	(2.16, 3.35)	0.279
**IL-6**	1.55 ± 1.22	5.09 ± 6.54	(0.94, 2.16)	(2.26, 7.92)	0.0294 *
**IL-8**	2.21 ± 1.55	4.70 ± 2.84	(1.44, 2.98)	(3.47, 5.93)	0.0019 **
**IL-12p70**	2.91 ± 0.75	3.78 ± 5.89	(2.54, 3.29)	(1.29, 6.27)	0.54
**IL-17**	1.01 ± 0.31	1.02 ± 0.32	(0.85, 1.17)	(0.87, 1.16)	0.947
**GM-CSF**	2.33 ± 0.26	2.58 ± 1.10	(2.20, 2.47)	(2.11, 3.06)	0.349
**IFN-γ**	3.05 ± 0.97	4.39 ± 1.96	(2.49, 3.61)	(3.44, 5.34)	0.0261 *
**TNF-α**	9.12 ± 1.91	14.97 ± 8.26	(8.17, 10.07)	(11.39, 18.54)	0.00556 **
**MCP-1 (CCL2)**	95.64 ± 55	274.60 ± 147.2	(68.29, 123)	(212.5, 336.8)	<0.0001 ****
**MIP-1α (CCL3)**	2.60 ± 1.03	3.79 ± 2.02	(2.077 3.14)	(2.89, 4.69)	0.0341 *
**MIP-1β (CCL4)**	13.02 ± 8.09	19.20 ± 10.80	(8.99, 17.04)	(14.64, 23.76)	0.0486 *
** *Anti-inflammatory* **					
**IL-4**	0.77 ± 0.17	0.81 ± 0.48	(0.67, 0.86)	(0.59, 1.032)	0.749
**IL-5**	0.67 ± 0.33	0.49 ± 0.20	(0.51, 0.84)	(0.40, 0.58)	0.0412 *
**IL-9**	5.04 ± 1.65	4.27 ± 2.14	(4.22, 5.86)	(3.34, 5.20)	0.215
**IL-10**	8.14 ± 3.16	11.36 ± 5.59	(6.56, 9.71)	(8.87, 13.84)	0.0365 *
**IL-13**	20.11 ± 20.52	22.62 ± 20.12	(−1.42, 41.65)	(−2.35, 47.60)	0.843

## Data Availability

The original data presented in the study are available on request from the corresponding author.

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
