# Peer review of "Plasma Microbiome in COVID-19 Subjects: An Indicator of Gut Barrier Defects and Dysbiosis"

_ijms, 2022, doi:10.3390/ijms23169141_

Round 1

Reviewer 1 Report

In this paper, the authors report blood dysbiosis (16S sequencing) and enhanced gut barrier leakage in COVID-19 patients. This is a novel and very interesting study. The experiment plan is well conducted and there are no major pitfalls. The litterature in the field is quite scarce at the moment and data shown are quite solid. Below are some comments that might ameliorate the quality of the study.

1-Correlation analyses

In general, this manuscript lacks correlation analyses. The authors should investigate whether there is a link (in COVID-19 patients)  between the nature of blood bacteria and systemic factors such as inflammatory cytokines (+ CRP and procalcitonin) and gut barrier leakage markers.

The authors should also classify/stratify the severity of COVID-19 and investigate whether for instance the bacteria blood profile of severe patients is different from those of mild or moderate patients. I don’t know if the number of patients will be sufficient but the authors should try. If the number is insufficcient, please discuss this point as a limitation. In general, limitations of the study should be discussed (see below).

Regarding the 16S analysis, one can see that the number of patients is lower relative to patients for whom blood was analyzed by ELISA. It would be interesting to show the ELISA data for patients analyzed by 16S (the current  Figure 4 can remain of course). In other words, is there a correlation between blood microbiota changes and alteration of gut barrier markers ? This is very important.

Regarding co-morbidities, have the authors found a higher blood dysbiosis and leaky gut in diabetic patients (if I understand well, approx 30% of the patients analyzed)?

Any correlation between in-hospital mortality, lenght of hospitalization and recovery period ? The authors should have this information.

2-Discussion

The discussion is not enough focused on data. This is no need to convince the reader regarding the interest of studying blood microbiota. The authors should rather concentrate on their data in the light of the current litterature.

-The authors should quote the venzon et al’ s paper (bioRxiv. 2022, doi: 10.1101/2021.07.15.452246) which is in line with the current observation (blood dysbiosis and enhanced gut barrier leakage in COVID-19 patients).

-To substantiate their findings, the authors should discuss studies reporting gut dysbiosis and gut barrier leakage in animal models of COVID-19, in particular in SARS-CoV-2-infected hamsters. The later is a suitable model of COVID-19, including long COVID-19 (according to Frere et al (2022). doi: 10.1126/scitranslmed.abq3059). Indeed, Sencio et al (2022) (doi: 10.1080/19490976.2021.2018900) reported a trend towards  enhanced level of iFABP in the blood of infected hamsters. This is in line with the present study.

The authors claim that gut dysbiosis/leaky gut may participate in secondary bacterial infection post-SARS-CoV-2 infection. Is there any paper showing that gut dysbiosis during a respiratory viral infection can lead to secondary bacterial infection ? If so, papers should be quoted . Here too, the aim is to find arguments to support the authors’s hypothesis.

3-Other comments/questions

It is not clear how many patients and healthy donors were analyzed (16S sequencing). It is mentioned 31 subjects (Covid + healthy ??), line 107, page 4,  but one can see more samples in Figure 1. Why only « 31 subjects » analyzed (the authors have access to 146 patient). A matter of cost ?

Please precise in the text when blood samples were collected. How long after hospitalization ?

There is a trend towards lower alpha diversity in COVID-19 patients (Fig1B and 1E).  This should be mentioned. Can the authors calculate the distance (Fig. 1F, beta diversity). Any clues regarding a Bray-Curtis analysis (same trend ?)

In general, no differences between male and females (cf Tables, Figures ?)?

 Regarding the presence of bacteria in healthy individuals, do the herein reported data fit with the literrature ? Is it something normal ? Some references indicating this is the case should be quoted (page 6).

How do the authors fully eliminate the possibility of contamination during blood sample collection ?

The concentration of LPS as well as PGN are increased in the blood of COVID-19 patients. If the authors have access to HEK-blue expressing TLRs, they might want to see whether this translates into HEK activation.

Author Response

Reviewer -1

In this paper, the authors report blood dysbiosis (16S sequencing) and enhanced gut barrier leakage in COVID-19 patients. This is a novel and very interesting study. The experiment plan is well conducted and there are no major pitfalls. The literature in the field is quite scarce at the moment and data shown are quite solid. Below are some comments that might ameliorate the quality of the study.

We highly appreciate the reviewer’s comment and valuable suggestions.

1-Correlation analyses

In general, this manuscript lacks correlation analyses. The authors should investigate whether there is a link (in COVID-19 patients) between the nature of blood bacteria and systemic factors such as inflammatory cytokines (+ CRP and procalcitonin) and gut barrier leakage markers.

We thank the reviewer for the insightful comment. Since the beginning, we tried to do our best, but we were unable to do this as the plasma sample for the patients in microbiome study was used entirely for the 16S analysis. We used a separate cohort for the other observations. Because of the rigor with which we obtained the samples from sterile central lines or peripheral lines there was a limitation on sample availability. Under certain conditions, we could achieve only a small amount plasma (250 µL/ COVID-19 patient). To perform the microbiome study, a minimum of 250 µL sample was required. We have discussed this limitation in the revised version of manuscript.

The authors should also classify/stratify the severity of COVID-19 and investigate whether for instance the bacteria blood profile of severe patients is different from those of mild or moderate patients. I do not know if the number of patients will be sufficient, but the authors should try. If the number is insufficient, please discuss this point as a limitation. In general, limitations of the study should be discussed (see below).

Our result indicates that 6-7 out of ten moderate COVID-19 patients (Table-4) show greater phylogenic differences compared to the mean. (Fig. 2B-E). However, our sample size for COVID-19 patients is too small to draw any conclusions and this is now discussed in the study limitation section.

Regarding the 16S analysis, one can see that the number of patients is lower relative to patients for whom blood was analyzed by ELISA. It would be interesting to show the ELISA data for patients analyzed by 16S (the current  Figure 4 can remain of course). In other words, is there a correlation between blood microbiota changes and alteration of gut barrier markers? This is very important.

We appreciate reviewer’s comment. Although our initial intention was to compare plasma microbiome with gut barrier markers, this could not be perform due to limitations of the sample.

Regarding co-morbidities, have the authors found a higher blood dysbiosis and leaky gut in diabetic patients (if I understand well, approx. 30% of the patients analyzed)?

In the current study, we could not compare the plasma microbiome with markers of leaky gut. The outcomes in diabetic patients are shown in Table 4 and of interest a lengthy hospitalization for up to 123 days was seen in one diabetic subject. Previously, we showed that diabetic individuals have more gut leakage than normal nondiabetic individuals (significantly higher levels of PGN and FABP2 in the plasma) (PMID: 31610731). Increased gut leakage in diabetic Covid-19 patients could contribute to the worse outcomes as reported by other groups (PMID: 32687793,34721437,33431578)

Any correlation between in-hospital mortality, length of hospitalization and recovery period? The authors should have this information.

In the updated Table 4 (Characteristics of COVID-19 patients used in measurement of the plasma microbiome), we include length of hospitalization and mortality which is increased with the severity of covid-19. Diabetes is one of the major risk factors for Covid-19 severity and extended hospitalization up to 123 days was observed in one of our diabetic subjects (Table 4).

 2-Discussion

The discussion is not enough focused on data. This is no need to convince the reader regarding the interest of studying blood microbiota. The authors should concentrate on their data in the light of the current literature.

-The authors should quote the venzon et al’ s paper (bioRxiv. 2022, doi: 10.1101/2021.07.15.452246) which is in line with the current observation (blood dysbiosis and enhanced gut barrier leakage in COVID-19 patients).

 -To substantiate their findings, the authors should discuss studies reporting gut dysbiosis and gut barrier leakage in animal models of COVID-19, in SARS-CoV-2-infected hamsters. The latter is a suitable model of COVID-19, including long COVID-19 (according to Frere et al (2022). doi: 10.1126/scitranslmed.abq3059). Indeed, Sencio et al (2022) (doi: 10.1080/19490976.2021.2018900) reported a trend towards enhanced level of iFABP in the blood of infected hamsters. This is in line with the present study.

The authors claim that gut dysbiosis/leaky gut may participate in secondary bacterial infection post-SARS-CoV-2 infection. Is there any paper showing that gut dysbiosis during a respiratory viral infection can lead to secondary bacterial infection? If so, papers should be quoted. Here too, the aim is to find arguments to support the authors’ hypothesis (PMID: 32788292)

We appreciate reviewer’s suggestion and revise the discussion to include the Venzon et al and Frere et. al. as suggested by the reviewer.   

3-Other comments/questions

It is not clear how many patients and healthy donors were analyzed (16S sequencing). It is mentioned 31 subjects (Covid + healthy ??), line 107, page 4,  but one can see more samples in Figure 1. Why only « 31 subjects » analyzed (the authors have access to 146 patient). A matter of cost ?

We appreciate reviewer’s comment. A total 36 (15 COVID-19, and 21 healthy individuals) were included in the plasma microbiome assessments. Information has been updated in the revised manuscript. Regarding sample size in each cohort, the cost of running the microbiome analysis was another issue limiting the number of samples that were run.

Please precise in the text when blood samples were collected. How long after hospitalization?

Information have been updated in the revised manuscript. Blood samples from the COVID-19 patients were collected during the first 48 hours of their hospitalization.

There is a trend towards lower alpha diversity in COVID-19 patients (Fig1B and 1E). This should be mentioned. Can the authors calculate the distance (Fig. 1F, beta diversity). Any clues regarding a Bray-Curtis analysis (same trend ?)

Unfortunately, we did not perform the Bray-Curtis analysis.

 In general, no differences between male and females (cf Tables, Figures?)?

In the revised text we include gender in the tables.

Regarding the presence of bacteria in healthy individuals, do the herein reported data fit with the literature? Is it something normal? Some references indicating this is the case should be quoted (page 6).

The reviewer brings up an excellent point and recently at paper by Deng et al. Cell Discovery 2021(PMID: 33750767) identified an intraocular microbiota in eye fluid of healthy individuals as the current dogma in ophthalmology and vision research presumes the intraocular environment to be sterile. However, recent evidence of intestinal bacterial translocation into the bloodstream and many other internal organs including the eyes, found in healthy and diseased animal models, suggests that the intraocular cavity may also be inhabited by a microbial community. Deng et al tested intraocular samples from over 1000 human eyes. Using quantitative PCR, negative staining transmission electron microscopy, direct culture, and high-throughput sequencing technologies, they demonstrated the presence of intraocular bacteria. The possibility that the microbiome from these low-biomass communities could be a contamination from other tissues and reagents was carefully evaluated and excluded. Furthermore, they revealed the presence of an intraocular microbiome in normal eyes from non-human mammals and demonstrated that this varied across species (rat, rabbit, pig, and macaque) and was established after birth. Additional citations have been added to results section.

How do the authors fully eliminate the possibility of contamination during blood sample collection?

The samples were collected from central lines and peripheral lines with appropriate sterile technique.

The concentration of LPS as well as PGN are increased in the blood of COVID-19 patients. If the authors have access to HEK-blue expressing TLRs, they might want to see whether this translates into HEK activation.

We appreciate reviewer’s suggestion. At present we do not have HEK activation data for these samples.

Reviewer 2 Report

Dear Authors, your manuscript titled "Plasma microbiome in COVID-19 subjects: an indicator of gut barrier defects and dysbiosis" is interesting, particularly at this time when the pandemic is still ongoing. Plasma microbiome studies are limited, and your research provides interesting and useful data for the scientific community.

It would be useful to implement in the introduction information on the microbial composition of plasma at least in healthy conditions.

You yourself have stressed the limits of your study, but I think they can be a point that can be resolved in the light of the data you have obtained.

Do you think that your results, in particular for the part of the intestinal mediators can lead to practical prognostic use of these markers? if yes please provide a comment in the discussion or conclusions.

In the text and in the references list please check and replace different typefaces in addition to their size.

Author Response

Reviewer-2

Dear Authors, your manuscript titled "Plasma microbiome in COVID-19 subjects: an indicator of gut barrier defects and dysbiosis" is interesting, particularly at this time when the pandemic is still ongoing. Plasma microbiome studies are limited, and your research provides interesting and useful data for the scientific community. It would be useful to implement in the introduction information on the microbial composition of plasma at least in healthy conditions. You yourself have stressed the limits of your study, but I think they can be a point that can be resolved in the light of the data you have obtained. Do you think that your results, in particular for the part of the intestinal mediators can lead to practical prognostic use of these markers? if yes please provide a comment in the discussion or conclusions.

We highly appreciate this reviewers’ encouraging comment regarding our study. The manuscript has been revised and includes information in the discission about the microbial population in healthy individuals.  We state: “Studies on the plasma microbiome are limited; however, Whittle et. al. performed a comprehensive evaluation of the blood microbiome in healthy and asthmatic individuals and found, at the phylum level, the blood microbiome was composed of Proteobacteria, Actinobacteria, Firmicutes, and Bacteroidetes.”

In the text and in the references list please check and replace different typefaces in addition to their size.

We thank the reviewer for bringing this to our attention and appropriate changes have been made in the revised manuscript.
